# Design and Implementation of a Generalized Safety Fault Diagnosis System for China Space Station Scientific Experimental Rack

**DOI:** 10.3390/s24165102

**Published:** 2024-08-06

**Authors:** Yifeng Wang, Tianji Zou, Lin Guo, Chenchen Zhang, Lu Zhang

**Affiliations:** 1Technology and Engineering Center for Space Utilization, Chinese Academy of Sciences, Beijing 100094, China; wangyifeng@csu.ac.cn (Y.W.); zoutianji@csu.ac.cn (T.Z.); guolin@csu.ac.cn (L.G.); zcc@csu.ac.cn (C.Z.); 2University of Chinese Academy of Sciences, Beijing 100049, China

**Keywords:** space station, space application, scientific experimental rack, safety, diagnosis

## Abstract

As astronauts stay in the China Space Station for a long time during the operation phase, how to ensure the long-term safety of the scientific experimental rack (SER) in the field of space application is a problem that needs to be solved urgently. Each SER in the field of space station applications is a complex system that faces risks from different hazards. At present, there is no generalized monitoring and diagnosis system for the common risks faced by the SER. In this paper, a generalized safety fault diagnosis system is proposed to ensure the long-term safe and stable work of SERs in orbit, considering the actual risks faced by the SER. With the design of a generalized main control board, a measurement and control board, and an SSPC (solid-state power controller) board, the software and hardware cooperate to realize the acquisition of various physical quantities, data processing, power supply and distribution management, and other functions. Combined with relevant fault detection algorithms, the real-time detection and diagnosis of the relevant risks, abnormality warnings, and fault disposal operations are realized, which can effectively ensure the safety of the payloads in the field of space application, astronauts, and the space-station system.

## 1. Introduction

With the successful launch of the Tianhe Core Module, the Wentian Lab Module, and the Mengtian Lab Module of the China Space Station (CSS), the in-orbit construction period of the CSS has been completed, and the in-orbit operation period of more than 15 years is about to begin. During the period of in-orbit operation, the space station will be utilized to carry out large-scale, continuous scientific and technological experiments and space application tasks involving multiple disciplines [1]. During the operation phase of the space station, astronauts will be on duty in orbit for a long period, and how to ensure the safety of astronauts and the space-station system during the operation and management phase is a prominent problem faced by the space station project. The experimental payloads of the application system in the operation phase of the space station mainly exist in the form of the SER, which has increased exponentially both in volume and complexity, and the amount of on-orbit data generated by them is far more than before, so it is not possible to efficiently monitor the status of various types of payloads by relying only on manual monitoring; at the same time, limited by the influence of satellite Tracking Telemetry and Command (TT&C) resources, the ground system alone cannot effectively deal with the sudden emergency risks in orbit. Therefore, it is necessary to design a generalized safety fault diagnosis system to realize all-around payload safety monitoring, to guarantee the safe conduct of scientific experiments, the safety of the equipment of scientific payloads, and the safety of astronauts and the space-station system.

## 2. Safety-Related Hazard Analysis of the Scientific Experimental Rack

During the construction and long-term operation of the space station, ensuring the safety of astronauts is the primary principle to be followed in the development of all manned spacecraft [2]. The SER of the CSS has the following characteristics which ensure the safety of the astronauts during the whole mission: ① The engineering technology of the SER is complex, with multiple supporting payloads and various system interfaces, which increases the probability of potential safety failures; ② this scientific experiment mission is complex in terms of profile, with many potential hazards, making safety assurance extremely challenging; ③ the astronauts will be deeply involved in the scientific experiment operations, which requires a high efficiency of cooperation between space and the ground to avoid potential safety risks caused by mistakes. The primary focus of the safety work of the SER is on various types of potential hazards. Therefore, the safety determination of the space station engineering SER focuses on the identification of hazardous sources [3]. The safety analysis of the SER is mainly conducted based on the three aspects: “general hazards, failure hazards, and medical hazards”. Among them, general hazards refers to the components or devices that may generate a high temperature, high pressure, and mechanical impact; failure hazards refers to the components or devices that may jeopardize the safety of the vehicle and the safety and health of the astronauts in the event of failure; medical hazards refers to the components or devices that may generate microbial contamination, noise, and ionizing radiation. The safety items identified are shown in the Figure 1 below.

Medical hazards are mainly eliminated and controlled during the ground development stage of the SER utilizing payload design, medical testing, and other methods, and this paper mainly focuses on the general hazards and fault hazards during the in-orbit operation of the SER. Different hazards have different effects on astronauts or spacecraft, and even in different cases of the same hazard, the consequences it generates can vary greatly depending on the different levels or concentrations, or when it is placed in different environments. Therefore, by using the PHA (preliminary hazard analysis) method, the important hazardous sources, and their hazardous characteristics, of the scientific experimental payloads in the operation phase of the space station are identified and determined as follows: (1) leakage of the liquid substance ethylene glycol, which is slightly toxic and can cause damages to the astronauts’ health and electronic devices; (2) an excessively high temperature of the payloads can lead to the deformation of the cold-plate or pipeline, which can lead to fire in serious cases; (3) short-circuiting and overcurrent of the electric circuit, which can also lead to fire in serious cases; (4) pressure abnormality of the pressure vessel during vacuuming, exhausting, filling with nitrogen, etc., and the rupture of vessels can lead to the loss of oxygen in the cabin, which can jeopardize the health of the astronauts, etc.; (5) abnormal operation of the fan in the thermal control system results in abnormal heat dissipation of the payload; (6) the failure of important devices related to the safety of the scientific payloads themselves. Comprehensively and accurately identifying all kinds of hazards of the SER, effectively combining the safety design of the SER with in-orbit disposal and other measures to eliminate hazards and control the hazardous consequences, and reasonably verifying and evaluating the safety are the focuses of the safety work of the SER.

## 3. System Design Requirements Analysis

### 3.1. Functional Requirements for Safety Diagnosis

Among the functional requirements for the SER are an acquisition and monitoring function for the thermal liquid flow, temperature, pressure, and other parameters in the SER, as well as fault detection, alarm, and processing capabilities.
The system must include a collection and monitoring function for the payload temperature of the SER, and have fault detection, alarm, and processing capabilities;The system must include a collection and monitoring function for the working voltage and (or) current of the payload of the SER, and have fault detection, alarm, and processing capabilities;The system must include a collecting and monitoring function for the internal pressure of the pressure vessel in the payload of the SER, and have fault detection, alarm, and processing capabilities;The system must include a collecting and monitoring function for the thermal control fan speed of the SER, and have fault detection, alarm, and processing capabilities;The system must include a unified number management, in-orbit measurement and control, and data transmission function for the payload of the SER, and have fault detection, alarm, and processing capabilities.

### 3.2. Performance Requirements for Safe Diagnosis

The performance requirements for safe diagnosis are as follows:The flow measurement range of the thermal liquid is 0~170 L/h, and the measurement accuracy is better than ±5%;The work pressure measurement range of the thermal liquid is 20~500 kPa, the static measurement accuracy is better than ±0.8% FS;The temperature measurement range of the thermal liquid is −10~60 °C, the accuracy is better than ±0.3 °C in the range of 0~40 °C, and better than ±1 °C in other ranges;The range of voltage acquisition in the analog telemetry data is 0~5 V, and the accuracy is better than ±50 mV;The pulse width of the internal OC programmed output of the system is 80 ± 10 ms;The time error of emergency power-off disposal is ≤1 s.

## 4. System Hardware Design Framework

The hardware composition of the system is shown in Figure 2.

The main control board processor adopts the Flash architecture of the SmartFusion2 series SoC chip, whose program storage Flash flip saturation cross-section can reach 10^−11^ orders of magnitude, together with the triple-mode redundancy of the key register parameters, the memory data EDAC, and the ECC checksum measures, which can effectively avoid the spatial single-particle events (e.g., SEU) leading to operational interruptions, as well as device function failure malfunctions [4].

The FPGA integrates a dual-core 166 MHz ARM^®^Cortex™-M3 [5] hard-core processor on-chip, with a processing speed that can meet the needs of SER digital management, digital transmission, power supply and distribution management, and fault monitoring and disposal tasks. The system fault status and information are summarized on the main control board for centralized judgment and feedback on disposal orders. The main control board and the scientific payload/measurement and control board are interconnected by an RS422 and 100/1000 Base-T network, constituting a dual-link heterogeneous backup interconnection system, and the link failure can be automatically switched to the idle link to continue communication, which can satisfy the hardware reliability and real-time requirements of fault response.

The solid-state power controller (SSPC) board isolates the driving circuit and controls the conduction or shutdown of the switch tube through the FPGA to realize the power distribution management and overcurrent and short-circuit protection of the scientific payload equipment, which has many advantages such as high reliability, flexible control, strong payload adaptability, easy-to-realize integration, intelligence, fault isolation, etc. [6]. The overcurrent and short-circuit threshold parameters can be changed on-rail to meet the dynamic power demands of different payloads. When an overcurrent fault occurs in the payload, the inverse time limit protection algorithm is activated to trigger the switching-off protection according to whether the integration result of the over-limit energy exceeds the threshold [7]; when short-circuit fault occurs in the payload, whether the fault current continuously meets the time window of the over limit is used as a criterion for controlling the switching-off protection, and the fastest response time is up to 15 microseconds, which can effectively prevent the occurrence of serious secondary faults such as cabin fires caused by payload short circuits and overcurrent faults.

The measurement and control board and scientific payload equipment, through the analysis of the main control data, are converted to analog control voltage, motor drive signals, I/O switching signals to drive the control of the fan, flow control valves, the entire rack stop valve, vacuum/exhaust valve control signals, etc., and through the A/D converter, 1-wire interface circuits, and I/O digital acquisition circuits to obtain the temperature, pressure, flow, smoke, and other safety-critical sensor data. The information is organized and sent back to the main control board for processing and comprehensive judgment. Measures such as strengthening the sensor selection and increasing the redundancy of measurement point channels are taken to ensure the authenticity and accuracy of the monitoring data, and to prevent the accidental triggering of faults caused by a single point failure of a certain sensor. For example, the environmental temperature sensor obtains data from various thermal control temperature collection points inside the load through a single measurement point dual-temperature sensor combined with a 1-wire dual bus temperature monitoring network, ensuring the authenticity and accuracy of key monitoring information.

## 5. System Software Function Realization

### 5.1. Software Interface Analysis

The software system interface is shown in Figure 3. The software system interface description is shown in Table 1.

### 5.2. Software Configuration Architecture

The software function module is shown in Figure 4, and is mainly composed of a communication management module, instruction processing module, and data processing module.

#### 5.2.1. Communication Management Module

The main control board communicates with the measurement and control board, SSPC board, and scientific payload through the RS422, all using the UART protocol; the baud rate is 115,200 ± 3% bps, and the communication is full-duplex, odd-check with 1 start bit, 1 stop bit, and 8 data bits.

The main control board can be interconnected with the measurement and control boards and the scientific payloads via a 100/1000Base-T network as a backup to the RS422 for digital tube communication, all using the TCP protocol.

The RS422 interface between the main control board and the SSPC board work as follows. The main control board sends data request instructions to the SSPC board through this interface according to the cycle, and the SSPC board sends data packets to the main control board through the interface within the agreed time after receiving the data request instructions, which include the working status of the SSPC board itself and the voltage, current, and power supply and distribution status corresponding to the collected scientific payloads. 

The RS422/Ethernet interface between the main control board and the measurement and control board works as follows. The main control board sends data request instructions to the measurement and control board through the interface periodically, and the measurement and control board sends data packets to the main control board through the interface within an agreed period after it receives the data request instructions, which include the measurement and control board’s working status and the collected information on various types of gas–liquid analog data, the status of the smoke sensors, the speed of the fans, the temperatures, and so on.

The RS422/Ethernet interface between the main control board and the scientific payload works as follows. The main control board sends data request instructions to the scientific payload through this interface on a cyclical basis, and the scientific payload receives the data request instructions and sends data packets to the main control board through this interface within an agreed period, including information about the payload’s status, in particular the payload’s health status parameter.

#### 5.2.2. Instruction Processing Module

The instruction processing module is divided into four sub-modules: fan-speed control processing, power supply and distribution instruction control processing, OC program control processing, and emergency message processing. When the instruction processing module receives the corresponding input, it distinguishes the instructions as belonging to one of the above sub-modules using “judgment words”, and then processes them by the corresponding procedures.

The fan speed control process is shown in Figure 5. When the instruction processing module receives the instruction of fan-speed control, the fan-speed control signal is generated through the fan control interface module, the chip select signal and data signal are output through the DA output module, and then the fan is controlled to reach the corresponding speed through the fan control DA. At the same time, the fan-speed acquisition module calculates the real-time speed of the fan based on the FPS signal, and the rotational speed of the fan is fed back to the data processing module through the fan control interface module.

The processing flow of the OC-programmed system is shown in Figure 6. When the instruction processing module receives the OC-programmed instruction, it generates a positive pulse of 80 ± 10 ms through the OC driver chip and external driver and controls the corresponding OC channel to send out the pulse through the register.

The processing flow of the power supply and distribution instructions is shown in Figure 7. When the instruction processing module receives an SSPC board power supply and distribution instruction, it forwards the instruction code to the SSPC board through the RS422 interface, and the SSPC board distinguishes the instruction according to the relevant parameters of the instruction code as belonging to the category of power supply interface on and off, power supply interface status restoration, or the updating of relevant parameters of the power supply and distribution interface of the scientific payload and makes the corresponding control command.

The processing flow of an emergency disposal message is shown in Figure 8. When the instruction processing module receives the emergency disposal message, the main control will immediately empty the event table and forward the emergency disposal message to each board and scientific payload in the system. Each board and scientific payload will notify the relevant parts and components to start the emergency disposal process after receiving the message. Then, the system will perform the power-off operation on the scientific payload after 98 s.

#### 5.2.3. Data Processing Module

The data processing module has a universal design. Considering the diversity of scientific payloads and the differences in the environments in which various types of risks are located, this system can configure parameters such as the payload’s power supply and distribution channels, reference current, short-circuit protection current, temperature sensor ID, gas pressure, liquid supply pressure, and temperature using data injection and parameter binding to realize the safety detection, abnormality alarms, and fault disposal of various types of payloads.

The systems time processing functions as follows. This system has a self-timekeeping function and a time-calibration function. It can receive an external input time code to complete the system time calibration, and at the same time, it can complete the following operations by itself through the relative time code of self-policing: (1) for the time stamp of engineering data; (2) for the timing of security detection, abnormality alarms, and fault disposal; (3) for the timed execution of the time code of the data injection—if the received time for the execution of the data injection is out of time by ≥3 s, then it will be discarded after timeout.

The systems engineering data processing functions as follows. The system groups and packages the engineering data of the internal boards, components, and collected scientific payloads, and downlinks them through the relevant links. The engineering data include the analog telemetry data collected by each sensor as well as the software working mode and working status of each board and component, as well as the setting value of the bookbinding parameters related to security detection.

The systems digital telemetry data processing functions as follows. The system can downlink the digital telemetry data of internal boards, components, and collected scientific payloads in real time. The digital telemetry data include information on the operating status, safety detection status, and alarm parameters of each component and scientific payload.

The systems power failure disposal functions as follows. When the system triggers emergency disposal in any working scenario (receiving emergency disposal message or detecting risk, triggering emergency disposal), it immediately suspends the current operation, starts the process of the disposal of power failure, immediately sends an emergency disposal message to the relevant payload, empties the event table at the same time, starts the corresponding timing, and realizes power outage disposal of the payload through the shutdown operation of each channel of the SSPC board after 98 ± 1 s.

The systems AD sensor acquisition functions as follows. The system selects the analog data to be acquired (liquid temperature, liquid flow, liquid pressure, gas temperature, gas pressure, smoke detection, etc.) through the GPIO to switch the multiplexed selector switch. The analog data acquisition cycle is ≤500 ms, the measurement range of the liquid temperature sensor is required to be from −10 °C to 60 °C, the measurement accuracy is required to be from 0 to 40 °C, which is higher than ±0.5 °C, and the rest is higher than ±1.2 °C; the liquid pressure sensor measuring range requirements are 20~500 kPa, and the static accuracy is higher than ±0.8% FS; the flow sensor single branch measuring range requirements are 0~120 L/h, and the accuracy is higher than ±8% FS. The analog data acquisition and processing mechanism is to collect one line of data every 30 ± 1 ms; the initial power supply analog switch is switched to the first channel, waits for 29.5 ms, and collects data continuously 16 times with an interval of 1 us between each collection. Then, the analog switch is switched to the second channel, waits for 29.5 ms, and collects the data of the second channel. Following the same logic, the system sequentially collects analog data from all 16 data channels. After completing data collection for the 16 data channels, the data of each channel are averaged to obtain the corresponding analog data.

The systems DS18B20 sensor acquisition management functions as follows. This system collects the temperature value of the DS18B20 temperature sensor on each payload and air duct through a 1-wire bus [8]; its patrol cycle is
≯12 s, the measurement range is required to be 10~110 °C, and the accuracy is higher than ±0.8 °C.

The systems DA output processing functions as follows. After receiving the control signal from the fan-speed control module, the DA output module sends the corresponding fan DA control signal, collects the FPS signal of the fan speed, calculates the fan speed, and returns it to the fan-speed control module. The fan control voltage regulation range is [2 V, 5 V], and the regulation accuracy is 0.5 V. The fan-speed control signal contains the fan drive trigger signal and fan speed. The FPS signal output from the fan is a square-wave signal with a frequency of 0~100 Hz. After sampling and processing the FPS signal, the number of rising and falling edges of the signal in 1 s is the number of fan revolutions in 1 s.

The systems SSPC data processing functions as follows. The system sets the reference current, short-circuit protection current, and other parameters of each scientific payload power supply and distribution channel through the data injection instruction, collects the real time power supply current of the payload as well as the peak value of the current after the current supply of the channel is powered on, and realizes the function of overcurrent protection in accordance with the I^2^ T protection curve; when overcurrent conditions are reached, the system automatically shuts down the power supply channel of the scientific payload and reports the status of abnormal shutdown through the digital quantity and the engineering data. The abnormal shutdown can be restored only through the instruction.

## 6. On-Orbit Fault Diagnosis Algorithm Implementation

Through the fault detection algorithm and different configuration parameters, this system can realize the safety risk detection, abnormality alarm, and fault disposal of many kinds of scientific payloads in orbit. It mainly recognizes leakage of the thermal liquid, a high temperature of the payload, the short circuit and overcurrent of the power supply circuit, abnormal pressure of the pressure vessel, abnormal speed of the fan, failure of payload device, and so on.

### 6.1. Leakage of Thermal Liquid: Simultaneous Pressure-Temperature Anomalies

The algorithm is characterized as follows. The total RTD (rack thermal drawer) outlet pressure (*P*) ≥ a kPa, any branch-returned liquid temperature (*T*_1_, *T*_2_, *T*_3_) ≥ b °C, and the duration of the state (t) reaches c s. (a, b, c are constants).
Anomaly Criteria: *P* ≥ a & (*T*_1_ ≥ b||*T*_2_ ≥ b||*T*_3_ ≥ b) & t ≥ c

### 6.2. Excessive Temperature Payloads: High Temperature of the Liquid at Inlet, High Temperature or Fire of the Payload or Air Duct Inside Rack


(1)High temperature of the liquid at inlet


The algorithm is characterized as follows. The total RTD inlet temperature (*T*_0_) ≥ a °C, the total RTD outlet temperature (*T*_4_) ≥ b °C, and the duration (t) of the state is up to c s (a, b, c are constants).
Abnormal Criteria: *T*_0_ ≥ a & *T*_4_ ≥ b & t ≥ c
(2)High temperature or fire of the payload or air duct inside rack

The temperature of any one of the 32 groups (64) of DS18B20 sensors on the payloads (*GZ*_1_*~GZ*_32_) collected by the RTD is ≥a °C, and the duration of this state (t) is up to b s; the payload power supply current (*i*) collected by the RCS (rack controller system) is ≥c or the payload power supply state of RCS (Z) is “d” (a, b, c, d is a constant).

The anomaly criteria are: (*GZ*_1_ ≥ a||*GZ*_2_ ≥ a…||*GZ*_32_ ≥ a) & t ≥ b & (*i* ≥ c||Z = “d”).
(a)The temperature of any one of the eight groups (16) of DS18B20 sensors on the air duct (*GF*_1_~*GF*_8_) collected by the RTD is ≥a °C and the duration of this state (t) reaches b s; the payload power supply current (*i*) collected by the RCS is ≥c or the payload power supply state of RCS (Z) is “d” (a, b, c, d are constants).

The anomaly criteria are: (*GF*_1_ ≥ a||*GF*_2_ ≥ a…||*GF*_8_ ≥ a) & t ≥ b & (*i* ≥ c||Z = “d”).
(b)The air inlet temperature (*T*_5_) of the RTD is ≥a °C and the smoke sensor (Y) of the RTD detects the presence of smoke (‘c’); the duration (t) of the state is up to b s (a, b, c are constants).

The abnormal criteria are: *T*_5_ ≥ a & Y = “c” & t ≥ b.

### 6.3. Short-Circuit, Over-Current, etc.

The algorithm is characterized as follows. When the current value and time collected by the corresponding channel of the SSPC board satisfy a fixed relationship, it is considered that there is a short circuit or overcurrent in the power supply of the circuit.

The abnormality criteria are: I^2^T, overcurrent protection threshold calculation formula: $$I_{b}^{c}\times K\times F\left(current\;sampling\;frequency\right)\times 2^{-4},$$
where *I_b_* is the reference current, *F* is the current sampling frequency; when the oversampling rate is 32, the sampling frequency is 625 KHz (C, K are constants).

### 6.4. Pressure Anomalies in Pressure Vessel


(1)When the differential pressure sensor used is a positive differential pressure sensor and negative pressure occurs in the pressure vessel, the differential pressure sensors (*P*_1_ and *P*_2_) directly taken by RCS are ≥a Pa. The number of consecutive groups (N) of this state reaches b, and the power supply and distribution status of payload (L) is “c” or “d” (a, b, c, d are constants).


The anomaly criteria are: *P*_1_ ≥ a & *P*_2_ ≥ a & N ≥ b & (L = c||L = d).
(2)When the differential pressure sensor used is a positive and negative differential pressure sensor (one sensor can collect negative and positive pressure at the same time), and negative pressure occurs in the pressure vessel, the differential pressure sensors directly collected by the RCS (*P*_1_ & *P*_2_) ≤ a Pa and the number of consecutive groups of this state (N) reaches b. The power supply and distribution status of payload (L) is “c “ or “d” (a, b, c, d are constants).

The anomaly criteria are: *P*_1_ ≤ a & *P*_2_ ≤ a & N ≥ b & (L = c||L = d).
(3)When overpressure occurs in the pressure vessel, the differential pressure sensors (*P*_1_ and *P*_2_) of the RCS direct mining are ≥a Pa. The number of consecutive groups (N) of this state reaches b, and the power supply and distribution status of payload (L) is “c” or “d” (a, b c, d are constants).

The anomaly criteria are: *P*_1_ ≥ a & *P*_2_ ≥ a & N ≥ b & (L = c||L = d).

### 6.5. Abnormal Operation of Fan

The algorithm is characterized as follows. When the RCS detects a fan speed (*N*) ≤ a and the duration of this state (t) reaches b s, then the power supply and distribution status of payload (L) is “c” or “d” (a, b, c, d are constants).

The anomaly criteria are: *N* ≤ a & t ≥ b & (L = c||L = d).

### 6.6. Failure of Critical Devices Related to the Safety of the Scientific Payload Itself


(1)When the failure level (*L*) = a||b||c of the payload health status parameter captured by the RCS and the number of consecutive groups (N) of this status reach d (a, b, c, d are constants), failure is considered to have occurred.


The anomaly criteria are: (*L* = a||*L* = b||*L* = c) & N ≥ d.


(2)When the failure level (*L*) = e of the payload health status parameter collected by the RCS and the number of consecutive groups (N) of this status reach d (e and d are constants), then failure is considered to have occurred.


The anomaly criteria are: *L* = e & N ≥ d.
(3)When the failure level (*L*) = f of the payload health status parameter captured by the RCS and the number of consecutive groups (N) of this status reach d (f and d are constants), then failure is considered to have occurred.

The anomaly criterion is: *L* = f & N ≥ d.

## 7. System Application and Analysis

Currently, the system in question has been implemented in the payload and ground mirror system of the China Space Station for an extended period. The subsequent discussion outlines the efficient identification and resolution of relevant system faults in three practical application scenarios.

In the first scenario, the system is capable of diagnosing “short circuits or overcurrent” fault. In Table 2, Section 4, Section 5 and Section 6 detail the current configuration parameters utilized during a load test employing SSPC channel 1, with a reference current setting of 5 A and a short circuit current setting of 10 A. Upon detection by the SSPC module that the channel’s maximum current has surged from 2.46 A to 11.17 A, the channel’s status is adjusted to 4 (indicating short circuit off), and the 100 V output voltage is deactivated.

In the second scenario, the system is monitoring and diagnosing “the pressure in a pressure vessel”. Figure 9 and Figure 10 show the pressure difference sensor readings of the vessel under a specific load during charging and venting, while Figure 11 and Figure 12 display the system’s simulated pressure difference readings. The system’s analog readings closely match the sensor readings, serving as a reliable indicator of the actual pressure. Additionally, the system ensures that the vessel’s internal pressure stays within a safe range during charging and venting, based on preset thresholds of −20 kPa and 0 Pa.

The third scenario illustrates the monitoring and diagnostic process of the system in relation to safety failures concerning “critical components of scientific payloads”. As indicated in Table 3, in the event of a failure within the payload component, it signals AAH via the designated status word. Subsequently, upon the system registering the health status word of the payload as “A” for a series of consecutive instances, it communicates the fault code A1H within the status word of the entire rack based on the relevant fault level. This fault code is then relayed through alarm parameters to alert higher-level equipment and ground personnel for intervention.

## 8. Conclusions

Since the operation of the China Space Station in orbit for more than 1100 days, this system has always carried out fault detection and early warning for the scientific experimental rack of the space application system, effectively avoiding in-orbit risks caused by liquid pipeline plugging, scientific payload exhaust, short circuit of the payload connector, payload device failure, and other operations and situations. Through joint fault analysis and reasonable disposal of all aspects within the system, the normal operation of the space utilization system payload of the China Space Station, the on-schedule conduct of scientific experiments (tests), and the output of scientific payoffs have been guaranteed.

## Figures and Tables

**Figure 1 sensors-24-05102-f001:**
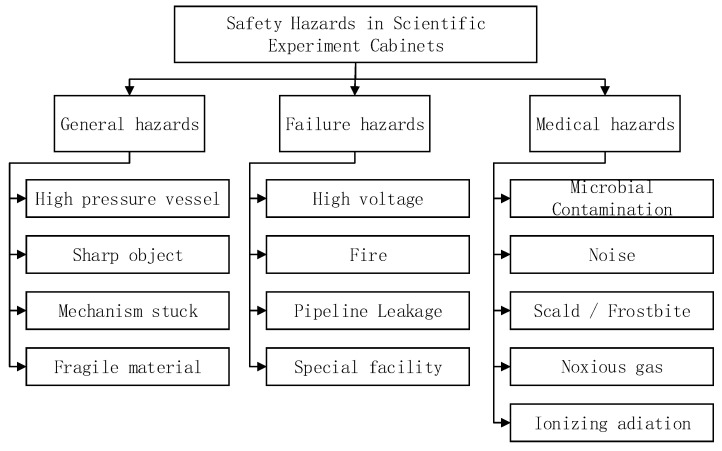
Safety-related hazard analysis of SER.

**Figure 2 sensors-24-05102-f002:**
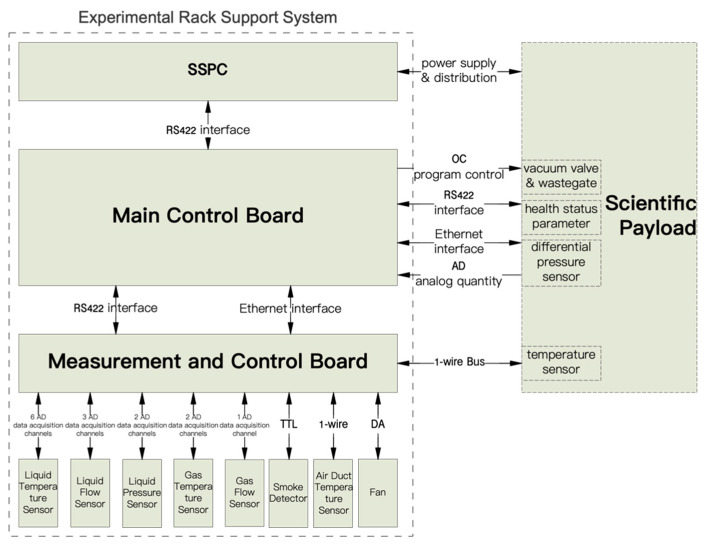
System hardware components.

**Figure 3 sensors-24-05102-f003:**
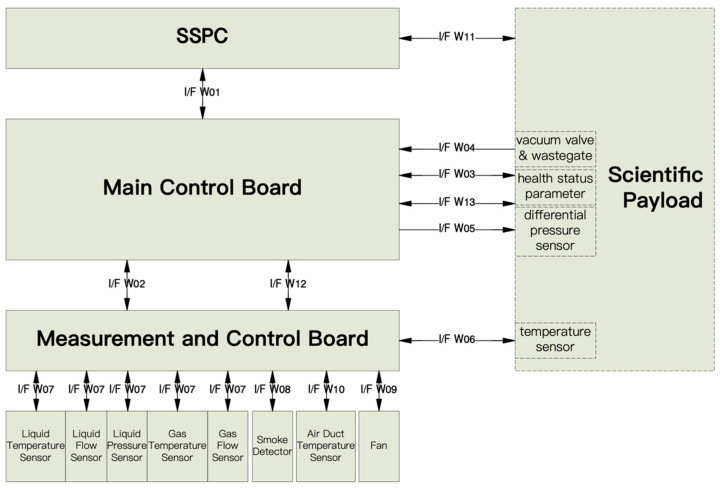
Software system interface.

**Figure 4 sensors-24-05102-f004:**
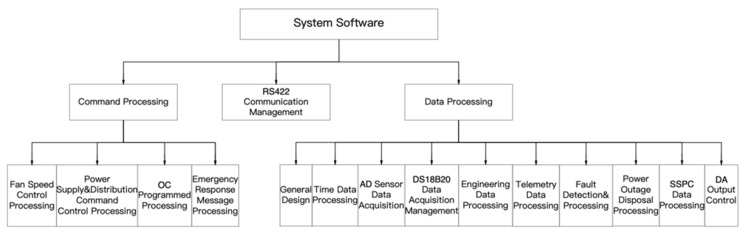
Software functional module.

**Figure 5 sensors-24-05102-f005:**
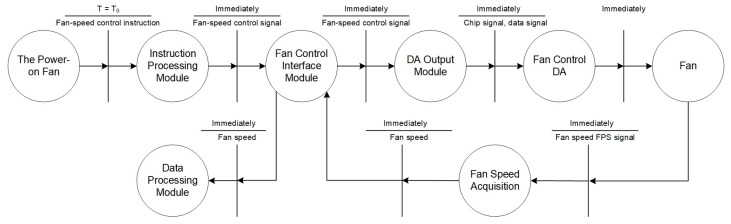
Fan-speed control process.

**Figure 6 sensors-24-05102-f006:**
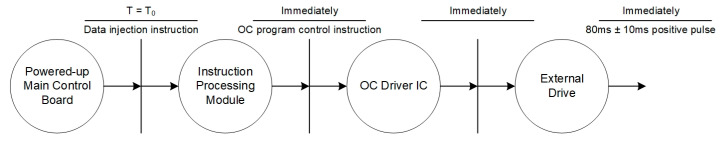
OC program control instruction process.

**Figure 7 sensors-24-05102-f007:**
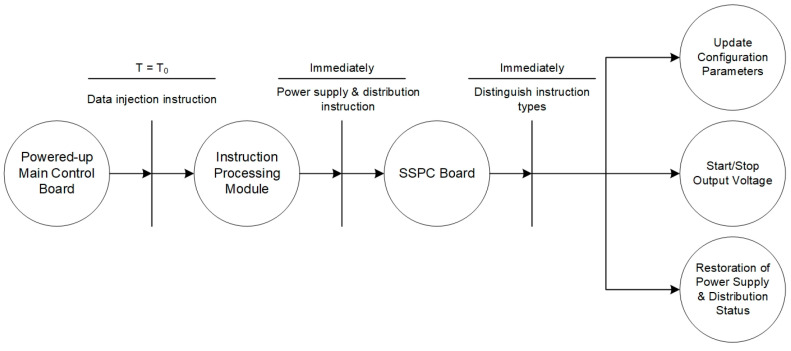
Power supply and distribution instruction process.

**Figure 8 sensors-24-05102-f008:**
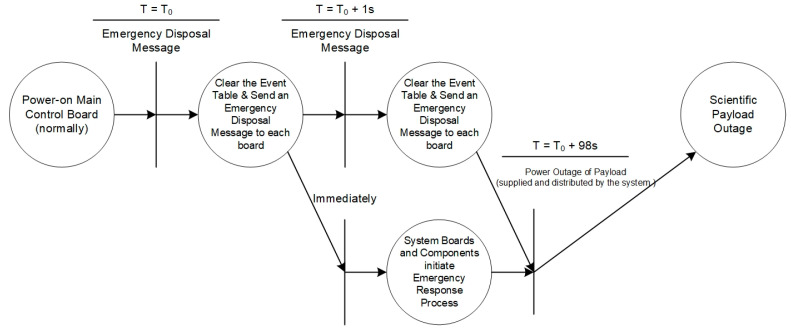
Emergency disposal message process.

**Figure 9 sensors-24-05102-f009:**
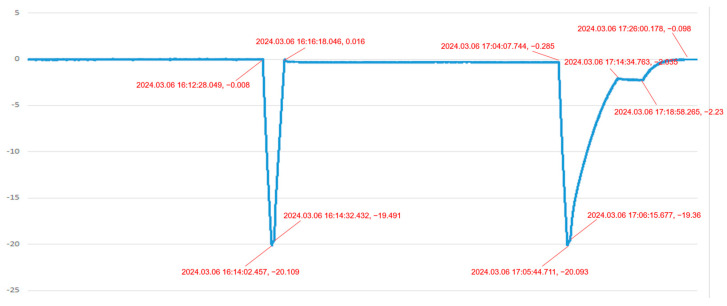
Pressure difference sensor 1 measurement curve of a pressure vessel under charging and venting condition.

**Figure 10 sensors-24-05102-f010:**
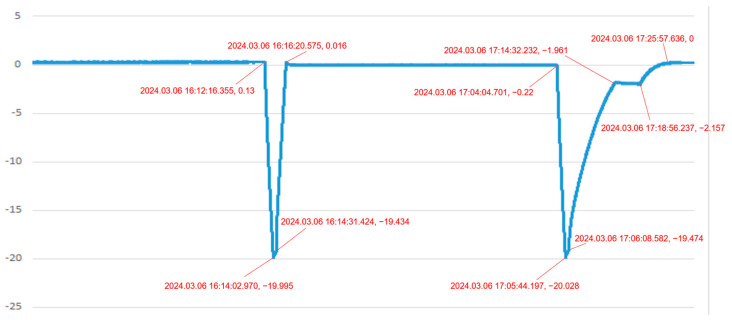
Pressure difference sensor 2 measurement curve of a pressure vessel under charging and venting condition.

**Figure 11 sensors-24-05102-f011:**
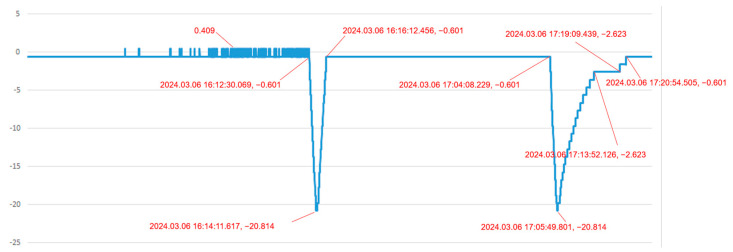
The system’s pressure difference analog data 1 of a pressure vessel under charging and venting condition.

**Figure 12 sensors-24-05102-f012:**
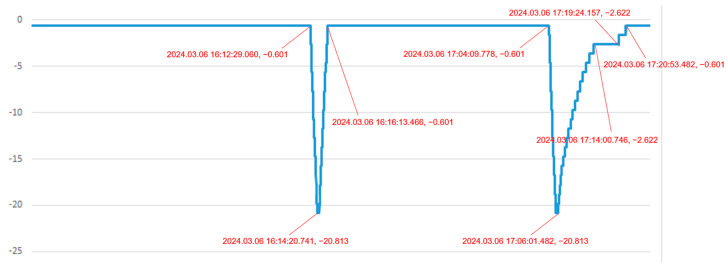
The system’s pressure difference analog data 2 of a pressure vessel under charging and venting condition.

**Table 1 sensors-24-05102-t001:** Software system interface description.

Interface Name	Interface Identification	Function Introduction
RS422 interface between main control board and the SSPC board	I/F W01	(1) The main control board receives the engineering data fed back from the SSPC board through this interface, which includes the payload voltage and current values collected by the SSPC board, as well as the abnormal status of the payload in case of overcurrent, short-circuit, and other abnormal phenomena; (2) The SSPC board receives instructions sent by the main control board through this interface and can supply and distribute 100 V and 28 V to the payload.
RS422 interface between main control board and measurement and control boards	I/F W02	(1) The main control board sends data injection instructions, etc. to the measurement and control board through this interface, and can control the speed of the fan in the thermal control system; (2) The measurement and control board sends digital telemetry data and engineering data to the main control board through this interface, which includes liquid temperature, flow rate, pressure, gas temperature, gas flow rate, smoke detector status, fan speed, and other information collected by the measurement and control board.
RS422 interface between the main control board and the scientific payloads	I/F W03	(1) The main control board sends data injection instructions, timecode and other messages to the scientific payload through this interface; (2) The scientific payload sends digital telemetry data and engineering data to the main control board through this interface, which contains the health status parameter.
OC programmable interface of the main control board to the scientific payloads	I/F W04	The main control board sends OC pulses to the scientific payload through this interface; the scientific payload can use this interface to control its internal related modules, for example, it can control the switch of the self-locking valves of the vacuum and exhaust lines through OC program control.
AD interface of the main control board to the scientific payloads	I/F W05	The main control board collects the corresponding analog data of the scientific payload through this interface, and then downlinks it through the digital telemetry data; the scientific payload can monitor its own status through the digital telemetry data of the main control board, for example, it can monitor the pressure situation of the internal environment of the scientific payload through this AD interface.
1-wire bus interface between the board and the scientific payloads	I/F W06	The measurement and control board can collect the temperature of the DS18B20 temperature sensor of the scientific payload through this interface.
AD interface between the board and each sensor	I/F W07	The measurement and control board can collect liquid temperature, liquid flow, liquid pressure, gas temperature, gas flow and other analog telemetry data through this interface.
Digital Pulse Signal Interface between Control Board and Smoke Detector	I/F W08	The interface allows the board to collect the smoke detected by the smoke detector.
DA chip control interface between the measurement and control board and the wind turbine	I/F W09	The measurement and control board can control the fan speed through the DA output (0~5 VDC) of this interface and collect the fan speed signal FPS through this interface.
1-wire bus interface between the measurement and control board and the air duct DS18B20	I/F W10	The measurement and control board can collect the temperature of the DS18B20 temperature sensor of the air duct through this interface.
Supply and distribution interface of the SSPC board to scientific payloads	I/F W11	The SSPC board provides 100 V and 28 V power supply to the scientific payload, and also collects the current operating voltage and current of the scientific payload.
Ethernet interface between the main control board and measurement and control boards	I/F W12	(1) Serves as a backup interface for I/F W02; (2) The main control board sends data injection instructions, etc. to the measurement and control board through this interface, and can control the speed of the fan in the thermal control system; (3) The measurement and control board sends digital telemetry data and engineering data to the main control board through this interface, which includes liquid temperature, flow rate, pressure, gas temperature, gas flow rate, smoke detector status, fan speed, and other information collected by the measurement and control board.
Ethernet interface between the main control board and the scientific payloads	I/F W13	(1) Serves as a backup interface for I/F W03; (2) The main control board sends data injection instructions, timecode and other messages to the scientific payload through this interface; (3) The scientific payload sends digital telemetry data and engineering data to the main control board through this interface, which contains the health status parameter.

**Table 2 sensors-24-05102-t002:** The occurrence of a current short circuit in the payload.

Time Code	100 V Maximum Current of Main Channel 1 (A)	100 V Main Channel 1 Status	SSPC Channel Name	SSPC Reference Current (A)	SSPC Short Circuit Current (A)
44:12.7	2.46	2	5	11.99	23.99
44:13.2	2.46	2	6	11.99	23.99
44:13.7	2.46	2	7	11.99	23.99
44:14.2	2.46	2	8	11.99	23.99
44:14.7	2.46	2	9	3	8.99
44:15.2	11.17	4	10	3	8.99
44:15.7	11.17	4	11	4	7.99
44:16.3	11.17	4	12	4	7.99
44:16.8	11.17	4	13	6	11.99
44:17.3	11.17	4	14	6	11.99
44:17.8	11.17	4	15	6	11.99
44:18.3	11.17	4	16	5	14.99
44:18.8	11.17	4	17	6	11.99
44:19.3	11.17	4	18	4	11.99
44:19.8	11.17	4	19	6	11.99
44:20.3	11.17	4	20	6	11.99
44:20.8	11.17	4	21	6	11.99
44:21.3	11.17	4	22	6	11.99
44:21.8	11.17	4	23	2	7.99
44:22.3	11.17	4	24	2	7.99
44:22.8	11.17	4	1	5	10
44:23.4	11.17	4	2	5	10
44:23.9	11.17	4	3	11.99	23.99
44:24.4	11.17	4	4	11.99	23.99

The highlighted content in the table represents the following meanings from left to right: the first column indicates the moment when the fault was diagnosed; the second column shows the maximum current value recorded at the time of diagnosis; the third column details the channel report after the fault has occurred, indicating that it is already in a short-circuit state; the fourth column specifies the channel number; the fifth column presents the reference current value configured for the channel; and the sixth column outlines the short-circuit protection current value set for the channel.

**Table 3 sensors-24-05102-t003:** The health status alarm of payload device failure.

Payload Time	Payload Health Status Word	Rack Time	Alarm Parameter	Rack Health Status Word
18:45:36.261	00H	18:45:38.585	0000H	00H
18:45:36.761	00H	18:45:39.094	0000H	00H
18:45:37.261	00H	18:45:39.585	0000H	00H
18:45:37.761	00H	18:45:40.106	0000H	00H
18:45:38.161	AAH	18:45:40.606	0000H	00H
18:45:38.661	AAH	18:45:41.126	0000H	00H
18:45:39.161	AAH	18:45:41.628	0000H	A1H
18:45:39.661	AAH	18:45:42.123	0000H	A1H
18:45:40.161	AAH	18:45:42.643	E30AH	A1H
18:45:40.661	AAH	18:45:43.144	E30AH	A1H
18:45:41.161	AAH	18:45:43.655	E30AH	A1H
18:45:41.661	AAH	18:45:44.155	E30AH	A1H
18:45:42.161	AAH	18:45:44.663	E30AH	A1H
18:45:42.561	AAH	18:45:45.163	E30AH	A1H
18:45:43.061	AAH	18:45:45.673	E30AH	A1H
18:45:40.161	AAH	18:45:42.643	E30AH	A1H

The highlighted sections in the table indicate the following meanings: the first two columns represent the time at which the payload reports an abnormality and the corresponding fault code value through the health status word. The subsequent three columns detail the faults identified after the fault diagnosis system conducts an analysis based on the fault codes reported by the payload, including the code value and alarm parameter value.

## Data Availability

Data available on request from the authors.

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
