# Peer review of "Design and Implementation of a Generalized Safety Fault Diagnosis System for China Space Station Scientific Experimental Rack"

_sensors, 2024, doi:10.3390/s24165102_

Round 1

Reviewer 1 Report

Comments and Suggestions for Authors

The conclusions do not confirm the scientific novelty of the article, which nullifies the results of reliable operation of programs and sensors during operation. There is no scientific novelty of the article in the abstract or in the conclusion. How the specific planned control was carried out, how the service life was carried out or laid down, did not require intervention or how they intervened was not taken into account. The article must be given a specific fact or process that made it possible to specify or write an article. Without this data, the article will look like a report to the management.

Reviewer 2 Report

Comments and Suggestions for Authors

This paper focuses on designing a universal fault diagnosis system to ensure the stable operation of SER. The study has important implications, however, this manuscript seems to be a product description, seriously lacking theoretical support and scientific verification. There are several important issues that need to be addressed in this paper.

1. The abstract of the manuscript needs major revision, and I do not see the scientific problems, theoretical problems, research status and the author's research basis of the study

2In the first or second part, you should show the content and focus of the research, preferably with clear diagrams. There are many inaccuracies and unclear contents in the article, such as the second part.

3. Is the font format in the first paragraph of Part 3 clearly different from other paragraphs in order to highlight some special meaning? Additionally, the numbering of the content in part 3.2 is repeated.

4. Figure 2 is not clear enough, for example, I can not find the FPGA in Figure 2 where.

5. Units of time on line 157 cannot be written this way.

6. The software and algorithm design part, namely the fifth and sixth parts, just like the function introduction, does not reflect the advanced nature of the algorithm and simple software design ideas.

7. The object of fault diagnosis in Part 6 is not clearly stated, and the logic is not clear.

8. Some abbreviations introduced in the paper have not been given their full names, and there is a lack of standardized definitions.

9. The most critical point is that this manuscript has not been verified by experiments, and I am not sure whether the content designed in the manuscript is feasible or can meet its performance requirements.

10. The references are too few and very old, and I do not think that they can help this manuscript to achieve a satisfactory result.

Comments on the Quality of English Language

Moderate editing of English language required

Reviewer 3 Report

Comments and Suggestions for Authors

SPECIFIC QUESTIONS

1.     What are the three modules of the China Space Station (CSS) that have been successfully launched?

2.     What is the expected duration of the CSS's in-orbit operation period?

3.     Why is it not possible to efficiently monitor the status of various types of payloads solely through manual monitoring during the CSS's operation phase?

4.     What necessitates the design of a generalized safety fault diagnosis system for the CSS?

5.     What are the primary goals of the generalized safety fault diagnosis system in ensuring safety during the CSS's operation phase?

6.     What is the primary principle in the development of all manned spacecraft during the CSS's construction and operation phases?

7.     List three characteristics of the Scientific Experimental Rack (SER) of CSS that contribute to ensuring astronaut safety.

8.     What are the three main types of hazards analyzed for the SER?

9.     How are medical hazards mainly controlled during the ground development stage of the SER?

10.  Identify and describe one example of each type of hazard (general, failure, and medical) associated with the SER.

11.  What are the functional requirements for safety diagnosis in the SER?

12.  What are the performance requirements for safe diagnosis regarding thermal liquid flow measurement?

13.  Describe the accuracy requirements for the work pressure measurement range of thermal liquid in the SER.

14.  What is the required time error for emergency power-off disposal in the SER?

15.  What processor architecture does the main control board of the SER's fault diagnosis system use?

16.  How does the system ensure reliability and real-time requirements for fault response in its hardware design?

17.  What is the role of the solid-state power controller (SSPC) board in the system?

18.  How does the system prevent serious secondary faults like cabin fires caused by payload short circuits and overcurrent faults?

19.  How does the system ensure the authenticity and accuracy of monitoring data from sensors?

20.  What is the communication protocol used between the main control board and other boards or scientific payloads in the SER system?

21.  Describe the process for fan speed control in the SER system software.

22.  What happens when the emergency disposal message is received by the instruction processing module?

23.  How does the data processing module handle the diversity of scientific payloads and their different risk environments?

24.  What is the measurement accuracy requirement for the liquid temperature sensor in the analog data acquisition process?

25.  What conditions must be met for the fault detection algorithm to identify a thermal liquid leakage in the SER?

GENERAL QUESTIONS

(i) Title of the manuscript could be further amended to reflect the novelty of the research conducted. As of now, it looks good but could be further improved to reflect the novelty.

(ii) An abstract is a shortened version of the work conducted and should contain all information necessary for the reader to determine. It would be great if the author could improve by adding those elements:

(1) what the objectives of the study were;

(2) how the study was done;

(3) what results were obtained;

(4) and the significance of the results.

(5) the application where the research could be implemented.

(iii) References as mentioned should focus more references between 2019 to 2024.

Comments on the Quality of English Language

SPECIFIC QUESTIONS

1.     What are the three modules of the China Space Station (CSS) that have been successfully launched?

2.     What is the expected duration of the CSS's in-orbit operation period?

3.     Why is it not possible to efficiently monitor the status of various types of payloads solely through manual monitoring during the CSS's operation phase?

4.     What necessitates the design of a generalized safety fault diagnosis system for the CSS?

5.     What are the primary goals of the generalized safety fault diagnosis system in ensuring safety during the CSS's operation phase?

6.     What is the primary principle in the development of all manned spacecraft during the CSS's construction and operation phases?

7.     List three characteristics of the Scientific Experimental Rack (SER) of CSS that contribute to ensuring astronaut safety.

8.     What are the three main types of hazards analyzed for the SER?

9.     How are medical hazards mainly controlled during the ground development stage of the SER?

10.  Identify and describe one example of each type of hazard (general, failure, and medical) associated with the SER.

11.  What are the functional requirements for safety diagnosis in the SER?

12.  What are the performance requirements for safe diagnosis regarding thermal liquid flow measurement?

13.  Describe the accuracy requirements for the work pressure measurement range of thermal liquid in the SER.

14.  What is the required time error for emergency power-off disposal in the SER?

15.  What processor architecture does the main control board of the SER's fault diagnosis system use?

16.  How does the system ensure reliability and real-time requirements for fault response in its hardware design?

17.  What is the role of the solid-state power controller (SSPC) board in the system?

18.  How does the system prevent serious secondary faults like cabin fires caused by payload short circuits and overcurrent faults?

19.  How does the system ensure the authenticity and accuracy of monitoring data from sensors?

20.  What is the communication protocol used between the main control board and other boards or scientific payloads in the SER system?

21.  Describe the process for fan speed control in the SER system software.

22.  What happens when the emergency disposal message is received by the instruction processing module?

23.  How does the data processing module handle the diversity of scientific payloads and their different risk environments?

24.  What is the measurement accuracy requirement for the liquid temperature sensor in the analog data acquisition process?

25.  What conditions must be met for the fault detection algorithm to identify a thermal liquid leakage in the SER?

GENERAL QUESTIONS

(i) Title of the manuscript could be further amended to reflect the novelty of the research conducted. As of now, it looks good but could be further improved to reflect the novelty.

(ii) An abstract is a shortened version of the work conducted and should contain all information necessary for the reader to determine. It would be great if the author could improve by adding those elements:

(1) what the objectives of the study were;

(2) how the study was done;

(3) what results were obtained;

(4) and the significance of the results.

(5) the application where the research could be implemented.

(iii) References as mentioned should focus more references between 2019 to 2024.

Round 2

Reviewer 1 Report

Comments and Suggestions for Authors

The comment was taken into account and reviewed by the authors. The article can be published, it would be nice to get a review from a specialist on this topic in the editorial office.

Reviewer 3 Report

Comments and Suggestions for Authors

All amendments had been carried out. Accepted for publication